# Effectiveness of COVID-19 Vaccination against Severe Symptoms and Death Among Geriatric Inpatients: A Retrospective Cohort Study in Macao

**DOI:** 10.3390/vaccines12080933

**Published:** 2024-08-21

**Authors:** Xiao Zhan Zhang, Phyllis Hio Hong Wong, Kai Seng Lai, Bo Yang, Menghuan Song, Junjun Li, Carolina Oi Lam Ung

**Affiliations:** 1Respiratory Medicine Department, Kiang Wu Hospital, Macao, China; zxiao801@gmail.com; 2State Key Laboratory of Quality Research in Chinese Medicine, Institute of Chinese Medical Sciences, University of Macau, Macao, China; mc36260@um.edu.mo (P.H.H.W.); menghuansong@um.edu.mo (M.S.); yc37536@um.edu.mo (J.L.); 3Emergency Department, Kiang Wu Hospital, Macao, China; laikaiseng@yahoo.com.hk; 4Diagnostic Imaging Department, Kiang Wu Hospital, Macao, China; kwkskgloria687@gmail.com; 5Department of Public Health and Medicinal Administration, Faculty of Health Sciences, University of Macau, Macao, China; 6Centre for Pharmaceutical Regulatory Sciences, University of Macau, Macao, China

**Keywords:** COVID-19 vaccines, SARS-CoV-2 variants, Omicron BF.7, vaccine effectiveness, mortality, vaccination timing, elderly, geriatrics

## Abstract

Monitoring the effectiveness of COVID-19 vaccination is critical for understanding if the vaccinated population, especially the elderly, is adequately protected from the emergence of new SARS-CoV-2 variants. This study aimed to investigate the effects of COVID-19 vaccination on the severity of symptoms and mortality in hospitalized geriatric patients during the Omicron BF.7 surge in Macao. Data from electronic health records and vaccination registry of inpatients aged 60 years or above admitted to Kiang Wu Hospital from 12 December 2022 to 12 March 2023 were retrospectively analyzed. The study involved 848 people, including 426 vaccinated and 422 unvaccinated individuals. The mean CXR scores (8.95 ± 9.49 vs. 11.41 ± 10.81, *p* < 0.001) and the mean MEWS scores (0.96 ± 2.01 vs. 1.49 ± 2.45, *p* < 0.001) were lower in the vaccinated group. By comparing the dose counts, no significant difference was seen in the odds of death. Based on the time of the last vaccination, 128 people were categorized as complete and 298 as incomplete vaccination. The complete vaccination group showed a 54% (95% CI 0.23–0.91) reduction in mortality risk (*p* = 0.026). The study findings not only reconfirm the effectiveness of COVID-19 vaccination but, more importantly, highlight the importance of vaccination timing to maximize vaccines’ protective effect.

## 1. Introduction

Since the World Health Organization (WHO) announced a Public Health Emergency and characterized it as a pandemic in early 2020, the coronavirus disease 2019 (COVID-19) has caused an accumulated death of over seven million worldwide as of 16 June 2024 [1]. At the same time, the pandemic has urged the COVID-19 vaccine development to become the fastest in history and allow the deployment of COVID-19 vaccination as one of the major strategies adopted worldwide to combat COVID-19. By December 2020, more than 200 COVID-19 vaccine candidates, who employed various approaches to the vaccine design to activate the immune system and induce immunity via different mechanisms, were in development [2]. 

For instance, the first COVID-19 vaccine approved by the Food and Drug Administration of the U.S. in August 2021 used messenger RNA (mRNA) to trigger the immune response to produce the necessary antibodies for protection against infection [3]. Early evidence showed that the mRNA COVID-19 vaccine was highly effective against hospital admissions within 6 months after being fully vaccinated [4]. The inactivated whole-virus vaccine is another widely offered COVID-19 vaccine globally [5], which has been shown to be effective against infection, hospital admission and even death [6].

Clinical studies supporting the expedited approval of the COVID-19 vaccines provide important but limited information on their effectiveness. In fact, studies have shown that vaccine effectiveness wanes over time [7]. Moreover, the emergence of new variants has raised concerns about the effectiveness of existing COVID-19 vaccines due to their potential immune-escape nature caused by mutation. Understanding vaccine protection amid the emergence of new variants is especially crucial for the elderly population, as evidence has shown that the high transmissibility of the COVID-19 variants imposes higher risks to the older population, which underscores the importance of continuous monitoring of vaccine effectiveness.

Macao, one of China’s cities located on the southeast coast of China with a population of 680,000, reported the first imported COVID-19 case on 22 January 2020. With imported COVID-19 cases on the rise since then, the city activated a lockdown, closed its borders, and implemented mandatory quarantine procedures in order to prevent a community outbreak [8]. Macao began a citywide voluntary COVID-19 vaccination program with the inactivated virus vaccine in February 2021, which was immediately followed by the vaccine option of mRNA vaccine available from March 2021 [9]. 

Vaccination, in addition to other stringent public health policies, has played a crucial role in maintaining a low death rate. As of June 2022, 16 months after the start of the COVID-19 vaccination program, Macao recorded less than 300 COVID-19-positive cases and zero deaths [10]. However, the tight community protection was weakened when the Omicron variants BA.5 triggered the first community outbreak on 18 June 2022, resulting in 1821 new cases, with six deaths [11]. Immediately, more stringent public health policies were implemented [12], which brought the spread of the virus under control by the end of July. No new cases were recorded until another wave of infected cases with the BA.5 and the newly emerged BF.7 Omicron variants in early December 2022 [13,14]. When the overall vaccination rate in Macao was over 90%, all the public health policies were lifted. Within 3 weeks, a surge in COVID-19 cases was seen, with over 70% of the Macao population infected by late December 2022 [15]. 

Whether the COVID-19 vaccination remains effective against severe symptoms and death over time among the elderly is particularly important for Macao, as people aged 60 or above account for 21% of the population [16]. According to an earlier local study, old age was associated with increased mortality and increased disease severity of COVID-19 infection [13]. Among the elderly who were vaccinated, about 90% of the older population opted for the inactivated virus vaccine [17]. However, local studies about the duration of the effect of COVID-19 vaccines on this vulnerable population are lacking, especially for those who require hospitalization. 

Our study aimed to investigate the effectiveness of COVID-19 vaccination on the severity of symptoms and survival in geriatric patients hospitalized during the surge of the BF.7 Omicron variant in Macao.

## 2. Materials and Methods

### 2.1. Study Design 

This was a retrospective, single-center, cohort study of geriatric inpatients at Kiang Wu Hospital using the data extracted from the electronic health records. Kiang Wu Hospital is one of the three major hospitals in Macao that provided treatment to over 50% of all hospital visiting patients. 

### 2.2. Study Period and Participants

Patients aged 60 years or above who tested positive for SARS-CoV-2, admitted to Kiang Wu Hospital between 12 December 2022 and 12 March 2023, were considered eligible. Among them, patients who were hospitalized for less than 24 h, hospitalized for other causes prior to the said period, or readmitted or transferred to other hospitals were excluded. The subjects were categorized into two cohorts, vaccinated or unvaccinated, based on their COVID-19 vaccination history. 

Public health policies against COVID-19 were lifted on 12 December 2022. Prior to this date, the majority of Macao residents had not been exposed to the SARS-CoV-2 due to the stringent public health policy in place. Therefore, it can be inferred that the immune responses exhibited by the study sample were mainly induced by vaccination. 

### 2.3. Data Collection

#### 2.3.1. Electronic Health Record

Data were obtained from the inpatient electronic health record system of Kiang Wu Hospital, which included demographic data (such as age and gender), clinical data (such as comorbidities, date of admission and discharge/death, length of stay in hospital, and the clinical outcome (improved or deceased)), and diagnosis of COVID-19 infection confirmed using RAT or RT-PCR via an oropharyngeal and nasopharyngeal swab, as well as the first chest X-ray score (CXR) and the Modified Early Warning Score (MEWS) taken upon admission to the hospital. 

#### 2.3.2. COVID-19 Vaccination Record

Vaccination details, including the date of vaccination, the type of vaccine, and the number of doses received, were obtained from the patients at admission, as shown in the government vaccination registry. 

As explained above, 12 December 2022 was set as the date of the first major COVID-19 outbreak in Macao. As shown in Table 1, considering that the COVID-19 vaccines usually take 2 weeks to become effective and remain most effective during the first three months after vaccination [18], a patient who received their first dose within 2–12 weeks prior to the outbreak (i.e., 19 September–28 November 2022) or had their second or any subsequent doses within 12 weeks before the outbreak (after 19 September 2022) was considered complete vaccination. Patients who received their first dose less than 2 weeks before the outbreak (after 28 November 2022), during the outbreak (12 December 2022–12 March 2023), or over 12 weeks prior to the outbreak (before 19 September 2022) or had their second or any subsequent doses over 12 weeks prior to the outbreak (before 19 September 2022) were considered incomplete vaccination.

### 2.4. Statistical Analysis

Numerical variables were presented as mean ± SD. Discrete variables were presented as percentages. The qualitative data were evaluated using Pearson’s chi-square test, while comparisons of quantitative variables between two groups were performed using an independent-sample *t*-test; results with a *p*-value < 0.05 were considered to be statistically significant. The crude odds ratios (cORs) were analyzed using binary logistic regression to estimate the effect of vaccination on the outcome. For the vaccination variables that showed significance, the Kaplan–Meier analysis was conducted to assess the outcome of the study groups. Cox proportional hazard models were used to adjust for covariates (age, sex, comorbidities) and determine variables predictive of survival. Hazard ratios were reported with 95% confidence intervals. All analyses were performed using IBM SPSS Statistics, version 27.

## 3. Results

As shown in Figure 1, this study identified 1236 hospitalized patients with the diagnosis of COVID-19 between 12 December 2022 and 12 March 2023, of whom 301 patients were excluded due to age under 60, and 87 patients were excluded as they were either admitted to hospital for less than 24 h, admitted to hospital due to other diseases prior COVID-19 period, or admitted due to readmission or transfer to other hospital.

As shown in Table 2, among the 848 patients included in this study, 449 (52.9%) were males, 399 (47.1%) were females, with a mean age of 82.38 ± 10.22 years, and all with a diagnosis of COVID-19 infection confirmed using RAT or RT- PCR via an oropharyngeal and nasopharyngeal swab. The majority of the included patients (*n* = 815, 96.1%) had at least one comorbidity (hypertension 57.4%, cerebrovascular disease 48.6%, diabetes 32.2%, tumor 17.2%, renal failure 16.7%, chronic respiratory disease 14.2%, and other diseases 50.9%). Of all, only 2.6% of patients were on immunotherapy.

Of the 848 patients, 426 (50.2%) were vaccinated and 422 (49.8%) were unvaccinated. The mean age of the vaccinated patients was 81.59 ± 10.15 years, while that for the unvaccinated group was 83.17 ± 10.25 years (*p* = 0.024). There were no significant differences in the mean length of hospital stay between the vaccinated group (13.84 ± 11.31 days) and the unvaccinated group (14.96 ± 11.71 days) (*p* = 0.159). A higher death rate was observed in the unvaccinated group (107; 25.4%) compared with the vaccinated group (64; 15.0%) (*p* < 0.001).

As shown in Table 3, the mean ± SD CXR score was 8.95 ± 9.49 and 11.41 ± 10.81 for the vaccinated and unvaccinated groups, respectively, and the number of patients having CXR scores over ≥20 was also significantly higher in the unvaccinated group (*p* < 0.001). The mean ± SD MEWS score was 0.96 ± 2.01 and 1.49 ± 2.45 for the vaccinated and unvaccinated group, respectively, and the number of patients having MEWS score ≥ 3 was also significantly higher in the unvaccinated group (*p* < 0.001).

As shown in Table 4, the crude odd ratio 1.92 (95% CI 1.36–2.71) of the unvaccinated group showed that the odds of death were nearly two times higher compared with the vaccinated group. Among the 426 vaccinated cases, 375 (88.0%) received the inactivated virus vaccine, 39 (9.2%) received the mRNA vaccine, and 12 (2.8%) received a mixed regimen. These patients had received one dose (*n* = 112, 26.3%), two doses (*n* = 142, 33.3%), three doses (*n* = 137, 32.2%), or four doses (*n* = 35, 8.2%) of COVID-19 vaccines. There was no statistically significant difference in their outcomes, whether it be improvement of symptoms or death, based on the vaccine type or the number of doses received. However, a significant difference was seen in the outcome when comparing the vaccination status. The crude odd ratio of 2.04 (95% CI 1.05–3.98) indicates that the odds of death in the incomplete vaccination group were two times higher compared with the complete vaccination group.

As there were significant differences when comparing the number of deaths between the vaccinated and unvaccinated groups (*p* < 0.001) and between the complete vaccination and incomplete vaccination groups (*p* = 0.035), further analysis was performed using Kaplan–Meier analysis adjusting for covariates.

Figure 2 shows the survival curves of the vaccinated and unvaccinated groups; a much steeper curve was seen in the unvaccinated group, which implies an increased probability in mortality. The divergence of the two curves also indicates there are difference in survival between the two groups. With a log-rank test result of *p* = 0.005, the better survival rate of the vaccinated group was statistically significant.

The Cox regression analysis revealed significance (*p* = 0.020) on survival, with the hazard ratio (HR) for the vaccinated group compared with the unvaccinated group was 0.68 (95% CI 0.49–0.94), indicating that the mortality risk in the vaccinated group was approximately 32% lower compared with the unvaccinated group. In contrast, older age (*p* = 0.005) and male sex (*p* = 0.008) showed significance in increasing mortality risk, with HR 1.02 (95% CI 1.01–1.04) and 1.56 (95% CI 1.12–2.16), respectively. Neither the use of immunosuppressants nor the number of comorbidities nor types of comorbidities had a statistically significant impact on the mortality risk. 

Figure 3 shows the survival curves of the complete and incomplete vaccination groups, although the curves did not diverge as much, the long horizontal line of the complete vaccination group commenced at around day 20, suggesting a longer survival duration with no further death observed. The log-rank test result was *p* = 0.052, so the difference between the group was not statistically significant. Both variables were further explored with Cox proportional hazard models to adjust for covariates.

The Cox regression analysis revealed significance (*p* = 0.022) on survival, with the hazard ratio (HR) for the complete vaccination group compared with the incomplete vaccination group being 0.45 (95% CI 0.22–0.89), indicating that the complete vaccination group had a 55% reduction in mortality risk. In contrast, male sex (*p* = 0.023), tumor (*p* = 0.043), and renal failure (*p* = 0.002) showed significance in increasing mortality risk, with HR 1.94 (95% CI 1.09–3.43), 2.55 (95% CI 1.03–6.31), and 4.66 (95% CI 1.75–12.41), respectively. Neither the use of immunosuppressants nor the number of comorbidities had a statistically significant impact on mortality risk. 

## 4. Discussion

This study found that COVID-19 vaccination, regardless of the type of vaccine, was beneficial in preventing severe cases of infection and improving survival outcomes when the new BF.7 variant was prevalent. Unlike the previous study, which selected a cohort within a specific timeframe of vaccine dose administration [6], this study included patients regardless of the time of their vaccine dose administration. Therefore, different time points for obtaining vaccination were taken into consideration when analyzing the vaccination effect. This allowed another important finding, which was that higher mortality risk was observed when the last vaccine dose was administered more than 12 weeks before the major outbreak. Collectively, the study findings not only reconfirm the effectiveness of the COVID-19 vaccines [19] but, more importantly, also highlight the importance of timely vaccination in order to achieve and maintain robust immunity [20].

A key difference in the findings of the current study was the impact of the number of vaccine doses on mortality rate. While previous studies found that additional vaccine doses were associated with increased effectiveness and reduced mortality [21,22], this study did not find a statistically significant difference in mortality rate among groups with different numbers of doses. 

Firstly, this could be due to the timing of vaccination. Within the group that received two doses of vaccines, over 70% of them had their last dose administered more than 12 weeks prior to the outbreak. Studies have shown that the level of antibodies after vaccination significantly declined 12 weeks after the second dose of vaccine, especially in the population aged over 60 [23]. As such, it could be deduced that the protective effect of two or more doses of vaccine might have diminished in the vaccinated group in this study. Considering that the vaccine doses were administered at different time points in this study, the potential benefits of additional vaccine doses might have been masked. Importantly, this draws attention to the timing of vaccination in addition to the number of vaccination doses. 

Secondly, due to the implementation of stringent quarantine policies, the Macao population had almost zero contact with any variants of COVID-19 prior to the lifting of the policy toward the end of December 2022. Therefore, their immunity was dependent on the effects of the vaccines, which could be seen in this study, where the effectiveness started to wane after 3 months of administration. In contrast to places like Hong Kong or other countries, where open policy was implemented much earlier into the start of the pandemic, other populations might carry natural immunity from previous undetected infections, potentially causing confounding effects to multiple vaccine doses. 

When comparing the timing of vaccination, although the study result did not show a clear, statistically significant difference in the overall survival curves between the complete and incomplete vaccination groups during the observed period, the *p*-value = 0.052 was very close to the significance threshold (*p* < 0.050). The small sample size and the number of censored subjects early in this study due to recovery may have contributed to the uncertainty [24], as the final outcome of the censored participants was unknown. Nevertheless, as the Kaplan–Meir analysis does not adjust for confounders, it was only considered as a first-level analysis in this study [25]. 

Further analysis was conducted, where age, sex, and other comorbidities were adjusted, and we observed a 55% reduction in mortality risk in the complete vaccination group. This finding is consistent with previous studies, which also suggested the antibody levels induced by the vaccines waned over time, an increased rate of COVID-19 infection was observed 3 months after the last vaccination on the Macao population [15], and the protection effect of vaccination decrease from the fourth month, especially in the older population [26]. 

Older patients are more vulnerable to COVID-19 due to immune dysfunction and elevated inflammatory markers associated with aging. Once infected, they often develop severe symptoms [27]. Gender was shown to be a risk factor when comparing the vaccinated and unvaccinated groups and the completed and incomplete vaccination groups. Being female was shown to be a protective factor with a lower mortality risk. This is consistent with a previous review study that suggested that female produces more antibodies and stronger immune response after vaccination [28,29]. 

The finding of this study did not identify cancer and renal failure as risk factors when comparing the vaccinated and unvaccinated groups. As over 70% of the patients in the study had their first dose of inactivated virus vaccine less than 2 weeks prior to the outbreak, they might not have sufficient antibodies for effective protection yet, as the immune response generally takes at least 2 weeks after vaccination to build up. Furthermore, as antibodies wane over time, the second dose of the inactivated vaccine should be administered at an interval of 3–4 weeks after the first dose, as recommended by the World Health Organization [30]. However, 16.8% of the patients in the study did not administer the second dose after 12 weeks, implying that their antibody levels might have waned off, resulting in no significant difference in the outcome when compared with the unvaccinated group. When examining the complete and incomplete vaccination groups, the results suggest that those with a diagnosis of cancer and renal failure were found to have over two times and four times greater mortality risk than those who did not. This is consistent with previous studies, which reported that cancer patients have been shown to have a lower humoral response to COVID-19 vaccines, resulting in lower sustained antibody levels compared with healthy populations [28]. Impaired renal function was reported to reduce immune response through different mechanisms [31]. The reduced estimated glomerular filtration rate (eGFR) in renal in renal failure patients was also shown to increase mortality due to infection [32]. Especially in patients undergoing dialysis, it was reported that the antibody response wanes more rapidly compared with healthy individuals [28]. Patients with other immunocompromised conditions, such as organ transplants or under treatment with an immunosuppressant, were reported to have a less effective immune response to COVID-19 vaccines, as they have lower antibody titers and a lower rate of seroconversion [33,34]. However, in this study, the mortality risks of the patients in these three subgroups (renal dialysis, kidney transplant, and immunotherapy) were not statistically significant; this could be limited by the small sample size of the subgroups. 

### 4.1. Implications for Vaccination Program in the Future

The findings from this study suggest several important considerations for vaccination programs in the future. As immune response takes time to develop and wane off over time, the timing of vaccine administration should also be considered. Therefore, when evaluating whether the population has acquired immunity against the disease, the number of vaccine doses could not be taken as the sole indicator. Although this study did not demonstrate a direct increase in mortality rate based on the time of vaccination, the significance of increased mortality risk represents an indirect indicator that should raise concern for enhancing and optimizing the vaccination program. 

### 4.2. Strengths and Limitations and Future Studies

One of the strengths of our study is that hospitalized patients receive standardized and consistent care and treatment, which can minimize interference. In addition, as Kiang Wu Hospital is one of the major hospitals in Macao, the data collected from the hospital in the study could be considered relatively representative. Furthermore, most of the data are based on inactivated virus vaccine, which is directly applicable to Macao, where inactivated virus vaccines were the preferred vaccine type in the older population.

This study also has several limitations. First, due to the limitations of the data source, information about lifestyle factors such as smoking status, demographic factors such as obesity, or environmental exposure was lacking. As such, the data analysis was conducted without capturing all the important underlying confounders. Second, the sample size of this study was relatively small, with some subgroups having a small number of patients. Third, this study period lasted for 3 months, and it was assumed that for all the included patients during this period, their immunity was solely a result of COVID-19 vaccination. The possibility of natural immunity was ruled out. Finally, this study did not follow up on long-term outcomes after patients’ discharge. 

Continuous studies are needed to monitor the protection of the vaccines against COVID-19 with regard to the new variants for different age groups. Instead of single-center data in this study, a multi-center study with a larger sample size and a more representative study sample will be needed to minimize statistical noise. A well-designed prospective study to better inform the data collection in the clinical setting and allow for more comprehensive data analysis is also warranted. Extended follow-up periods to assess long-term clinical outcomes after vaccination could also provide more insights into the durability of vaccine-induced protection. 

## 5. Conclusions

In conclusion, this study shows that COVID-19 vaccines provide protection against severe symptoms and death in the older population in Macao. More importantly, the study findings emphasized that not only the number of vaccine doses is important, but the timing of vaccine administration is also crucial to ensure the vaccine’s protective effect. Furthermore, individual variability and virus variants can affect the effectiveness of vaccines. In order to maximize protection, a dynamic vaccination strategy should incorporate the current and predictive epidemiological data to guide the timely expansion of vaccination coverage and the target recipient populations in future vaccination programs.

## Figures and Tables

**Figure 1 vaccines-12-00933-f001:**
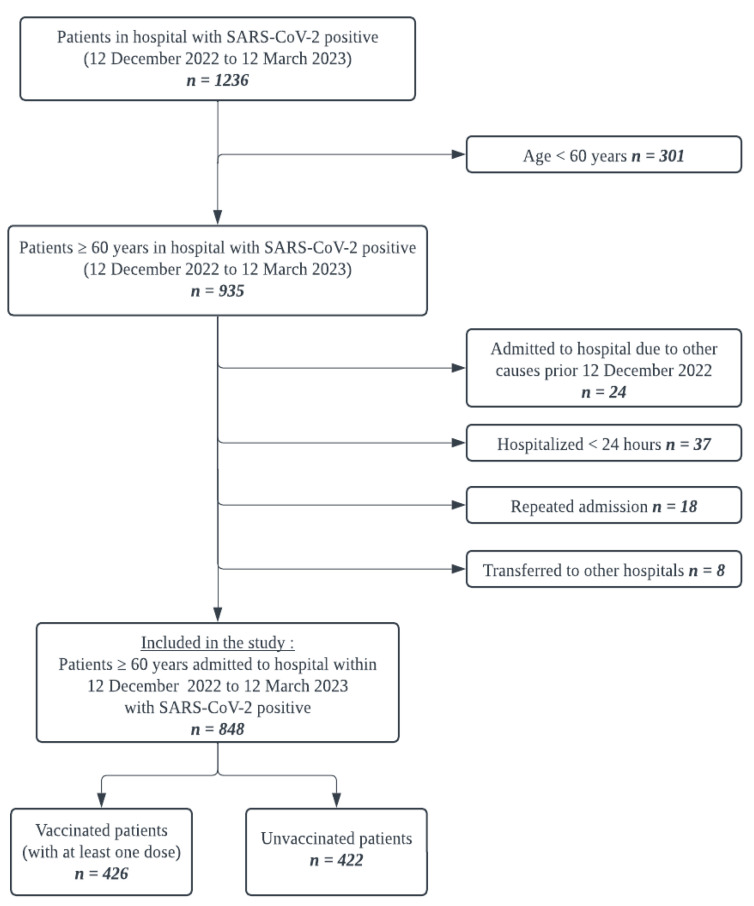
Flow diagram of participants in the study.

**Figure 2 vaccines-12-00933-f002:**
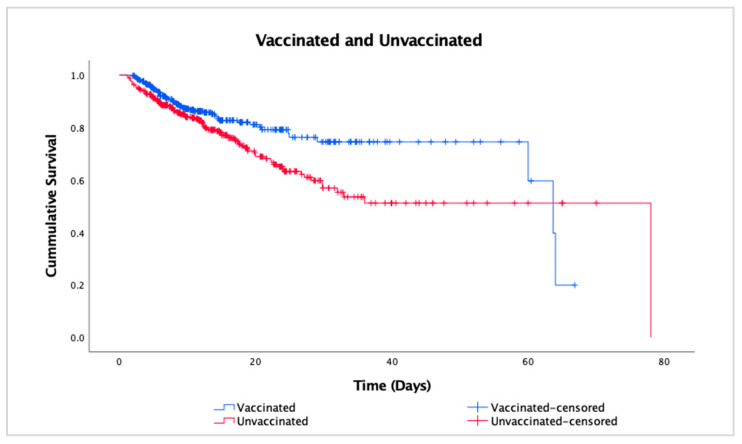
Survival curves for the vaccinated and unvaccinated groups.

**Figure 3 vaccines-12-00933-f003:**
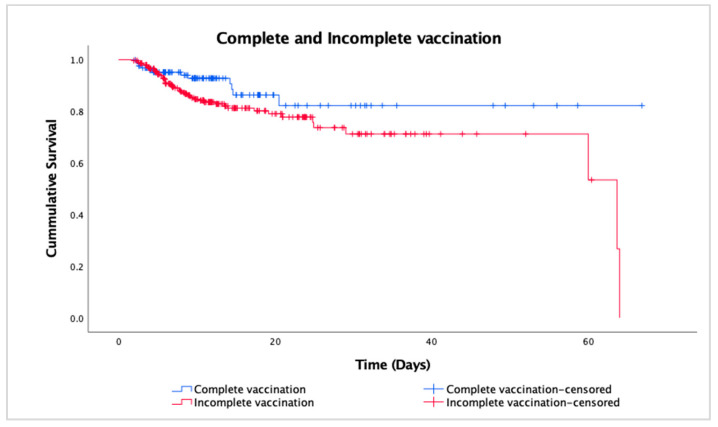
Survival curves for the completed and incomplete vaccinated groups.

**Table 1 vaccines-12-00933-t001:** Definition of complete vaccination and incomplete vaccination in this study.

	First Dose	Subsequent Doses
Complete vaccination	Within 2–12 weeks before the outbreak (19 September–28 November 2022)	Last dose < 12 weeks before the outbreak (after 19 September 2022)
Incomplete vaccination	<2 weeks before the outbreak (after 28 November 2022)	Last dose > 12 weeks before the outbreak (before 19 September 2022)
	During the outbreak (12 December 2022–12 March 2023)	
	>12 weeks before the outbreak (before 19 September 2022)	

**Table 2 vaccines-12-00933-t002:** Characteristics of the COVID-19 inpatients included in the study.

	All Patients	Vaccinated	Unvaccinated	t/x^2^	*p*-Value
	*n* = 848	*n* = 426 (50.2%)	*n* = 422 (49.8%)		
Demographics					
Age (years) (X ± SD)	82.38 ± 10.22	81.59 ± 10.15	83.17 ± 10.25	2.27	0.024
60–69 (*n*, %)	109 (12.9)	62 (14.6)	47 (11.1)		
70–79 (*n*, %)	242 (28.5)	122 (28.6)	120 (28.4)		
80–89 (*n*, %)	266 (31.4)	140 (32.9)	126 (29.9)		
≥90 (*n*, %)	231 (27.2)	102 (23.9)	129 (30.6)	5.96	0.114
Sex					
Female (*n*, %)	399 (47.1)	210 (49.3)	189 (44.8)		
Male (*n*, %)	449 (52.9)	216 (50.7)	233 (55.2)	1.73	0.188
Comorbidities					
Yes (*n*, %)	815 (96.1)	408 (95.8)	407 (96.4)		
No (*n*, %)	33 (3.9)	18 (4.2)	15 (3.6)	0.26	0.614
Number of comorbidities					
1 (*n*, %)	203 (23.9)	116 (27.2)	87 (20.6)		
2 (*n*, %)	248 (29.2)	125 (29.3)	123 (29.1)		
3 (*n*, %)	207 (24.4)	109 (25.6)	98 (23.2)		
≥4 (*n*, %)	157 (18.5)	88 (20.7)	69 (16.4)	7.04	0.071
Comorbidity types					
Hypertension (*n*, %)	487 (57.4)	226 (53.2)	261 (61.8)	6.52	0.011
Diabetes (*n*, %)	273 (32.2)	133 (31.2)	140 (33.2)	0.37	0.542
Malignant tumor (*n*, %)	146 (17.2)	73 (17.1)	73 (17.3)	0.00	0.950
Renal failure (*n*, %)	142 (16.7)	56 (13.1)	86 (20.4)	7.96	0.005
Renal dialysis	37 (4.4)	12 (2.8)	25 (5.9)	4.91	0.027
Kidney transplant	2 (0.2)	0	2 (0.5)	2.02	0.155
Cerebrovascular disease (*n*, %)	412 (48.6)	199 (46.8)	213 (50.5)	1.13	0.288
Chronic respiratory disease (*n*, %)	120 (14.2)	59 (13.8)	61 (14.5)	0.06	0.800
Other disease (*n*, %)	431 (50.9)	222 (52.1)	209 (49.8)	0.47	0.494
Immunotherapy (*n*, %)					
Yes	22 (2.6)	6 (1.4)	16 (3.8)		
No	826 (97.4)	420 (98.6)	406 (96.2)	4.76	0.029
Outcome					
Death (*n*, %)	171 (20.1)	64 (15.0)	107 (25.4)	14.06	<0.001
Length of hospital stay (days) (X ± SD)	14.40 ± 11.52	13.84 ± 11.31	14.96 ± 11.71	1.41	0.159

**Table 3 vaccines-12-00933-t003:** Comparison of clinical measurements of vaccinated and unvaccinated inpatients.

	All Patients	Vaccinated	Unvaccinated	t/x^2^	*p*-Value
	*n* = 848	*n* = 426 (50.2%)	*n* = 422 (49.8%)		
CXR Scores					
X ± SD	10.35 ± 11.03	8.95 ± 9.49	11.41 ± 10.81	3.49	<0.001
≥20 (*n*, %)	155 (18.3)	57 (13.4)	98 (23.2)	12.27	<0.001
MEWS Scores					
X ± SD	1.36 ± 3.85	0.96 ± 2.01	1.49 ± 2.45	3.44	<0.001
≥3 (*n*, %)	116 (13.7)	41 (9.6)	75 (17.8)	11.92	<0.001

**Table 4 vaccines-12-00933-t004:** Comparison of outcome based on vaccination status.

	Total (*n*, %)	Improvement (*n*, %)	Death (*n*, %)	cOR	95% C.I.	*p*-Value
Vaccinated						
Yes	426 (50.2)	362 (53.5)	64 (37.4)	Ref.	-	-
No	422 (49.8)	315 (46.5)	107 (62.6)	1.92	1.36–2.71	<0.001
Vaccine type						
Inactivated	375 (88.0)	318 (87.8)	57 (89.1)	Ref.	-	-
mRNA	39 (9.2)	32 (8.8)	7 (10.9)	1.22	0.51–2.90	0.652
Mix	12 (2.8)	12 (3.3)	0	0.00	0.00	0.999
Number of Doses						
1	112 (26.3)	95 (26.2)	17 (26.6)	Ref.	-	-
2	142 (33.3)	120 (33.1)	22 (34.4)	1.03	0.52–2.04	0.945
3	137 (32.2)	116 (32.0)	21 (32.8)	1.02	0.51–2.03	0.974
4	35 (8.2)	31 (8.6)	4 (6.3)	0.72	0.23–2.31	0.581
Vaccination Status						
Complete	128 (30.0)	116 (32.0)	12 (18.8)	Ref.	-	-
Incomplete	298 (70.0)	246 (68.0)	52 (81.3)	2.04	1.05–3.98	0.035

## Data Availability

The data presented in the study are available on request from the corresponding author.

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
