# Peer review of "Effectiveness of COVID-19 Vaccination against Severe Symptoms and Death Among Geriatric Inpatients: A Retrospective Cohort Study in Macao"

_vaccines, 2024, doi:10.3390/vaccines12080933_

Round 1

Reviewer 1 Report

Comments and Suggestions for Authors

The article it is a good written one, but if it is to see it from a critical point of view, i would suggest:

1. The title it is a bit unclear for the content of the manuscript.

2. Simplify the abstract for clarity and conciseness. Ensure it highlights the study's objective, methods, key findings, and conclusion.

3. Provide a better background on the importance of studying vaccine effectiveness against new variants.

4. It is a need for clearly description the study design, participants, and statistical methods. Ensure the explanation of data collection and analysis is detailed and straightforward.

5. Discuss the implications of your findings in the context of existing literature. Highlight the importance of timely vaccination and suggest areas for future research.

Reviewer 2 Report

Comments and Suggestions for Authors

This manuscript analyses the effectiveness of the COVID-19 vaccine against severe and fatal conditions during BF.7 Omicron subvariant among geriatric participants. The outcome focuses on geriatrics an area lacking in existing literature and is likely to interest readers.

Major concerns.

1. Did this study collect data linked to severe conditions, such as smoking status and obesity?

2. This study used a retrospective design. Why did this study not perform a binary logistic regression to determine the crude odds ratio (cOR) instead of Chi-square?

The OR outcomes are more meaningful than the Chi-square or t values, especially in the scope of vaccination effects.

Comments.

1. Suggest adding words, such as geriatrics, elderly or related to the Title and Keywords to make it consistent with the aims of this study.

2. In geriatrics, comorbidities are complicated.

Have you tried to classify the comorbidity count regardless of disease and tried to analyse it? (e.g. no, 1, 2, ≥3)

3. Table 2, Renal failure: Have you tried to classify renal replacement therapy and renal transplant, and analysed it?

4. Did some participants receive immunosuppressants? (e.g. organ transplants, autoimmune...)

Suggest clarifying it.

5. Table 4: Have you tried to classify the vaccine status by incomplete, complete and "booster"?

6. Significant figures: Suggest using the same significant figure of p-value throughout the manuscript. I found both 1, 2 and 3 digits. (e.g. <0.1, 0.03 and 0.005)

Typos.

1. Subscript â‚€, "Râ‚€". (line 71)

Round 2

Reviewer 2 Report

Comments and Suggestions for Authors

The authors have adequately addressed all the concerns I raised.

Comments.

1. Table 4: The crude odds ratios (cOR) for "Vaccination/Yes," "Vaccine type/Inactivated," "Number of Doses/1," and "Vaccination Status/Complete" should be reported as "Ref." (reference category), correct?

2. Table 4: The p-value of "Number of Doses/1". It was from binary logistic regression?
